# TP5, a Peptide Inhibitor of Aberrant and Hyperactive CDK5/p25: A Novel Therapeutic Approach against Glioblastoma

**DOI:** 10.3390/cancers12071935

**Published:** 2020-07-17

**Authors:** Emeline Tabouret, Herui Wang, Niranjana Amin, Jinkyu Jung, Romain Appay, Jing Cui, Qi Song, Antonio Cardone, Deric M. Park, Mark R. Gilbert, Harish Pant, Zhengping Zhuang

**Affiliations:** 1Neuro-Oncology Branch, Center for Cancer Research, National Cancer Institute, National Institutes of Health, Bethesda, MD 20892, USA; herui.wang@nih.gov (H.W.); jinkyu.jung@nih.gov (J.J.); jing.cui@nih.gov (J.C.); qisong@fudan.edu.cn (Q.S.); mark.gilbert@nih.gov (M.R.G.); 2Team 8 GlioME, CNRS, INP, Inst Neurophysiopathol, Aix-Marseille University, 13005 Marseille, France; romain.appay@ap-hm.fr; 3Neuronal Cytoskeletal Protein Regulation Section, National Institute of Neurological Disorders and Stroke, Bethesda, MD 20824, USA; aminn@ninds.nih.gov (N.A.); panth@ninds.nih.gov (H.P.); 4University of Maryland Institute for Advanced Computer Studies, College Park, MD 20742, USA; antonio.cardone@nih.gov; 5Department of Neurology and the Committee on Clinical Pharmacology and Pharmacogenomics, The University of Chicago, Chicago, IL 60637, USA; dpark9@neurology.bsd.uchicago.edu

**Keywords:** glioblastoma, CDK5, radiotherapy, chemotherapy, ATM, DNA damage, targeted therapy

## Abstract

We examined the efficacy of selective inhibition of cyclin-dependent kinase 5 (CDK5) in glioblastoma by TP5. We analyzed its impact in vitro on CDK5 expression and activity, cell survival, apoptosis and cell cycle. DNA damage was analyzed using the expression of γH2A.X and phosphorylated ATM. Its tolerance and efficacy were assessed on in vivo xenograft mouse models. We showed that TP5 decreased the activity but not the expression of CDK5 and p35. TP5 alone impaired cell viability and colony formation of glioblastoma cell lines and induced apoptosis. TP5 increased DNA damage by inhibiting the phosphorylation of ATM, leading to G1 arrest. Whereas CDK5 activity is increased by DNA-damaging agents such as temozolomide and irradiation, TP5 was synergistic with either temozolomide or irradiation due to an accumulation of DNA damage. Concomitant use of TP5 and either temozolomide or irradiation reduced the phosphorylation of ATM, increased DNA damage, and inhibited the G2/M arrest induced by temozolomide or irradiation. TP5 alone suppressed the tumor growth of orthotopic glioblastoma mouse model. The treatment was well tolerated. Finally, alone or in association with irradiation or temozolomide, TP5 prolonged mouse survival. TP5 alone or in association with temozolomide and radiotherapy is a promising therapeutic option for glioblastoma.

## 1. Introduction

Glioblastomas are the most frequent and aggressive primary brain tumors [1]. To date, no curative treatment is available despite improvements in surgical resection, radiotherapy and chemotherapy [2]. Recurrence occurs in a median time ranging from 6 to 10 months while median overall survival after optimal first line treatment is 14 to 16 months [2]. At relapse, response after cytotoxic chemotherapy is above 5%. Bevacizumab, a monoclonal antibody against VEGF, improved response rate up to 40% at recurrence but failed to improve overall survival in phase III trials [3,4,5]. So, the development of new therapeutic strategies is a critical unmet need in neuro-oncology. 

Recently, increasing evidence suggests a role of cyclin-dependent kinase 5 (CDK5) in cancer progression. CDK5 is an unconventional cyclin-dependent kinase which is highly expressed in brain compared to other organs [6]. Control of CDK5 activity is different from other CDKs. CDK5 is activated by binding to the non-cyclin activators p35 or p39, which are docked to cell membranes. In pathological conditions, such as cancer or neurodegenerative diseases, CDK5 cofactors are cleaved by calpain that removes the first 98 and 100 amino acids, producing p25 and p29, respectively. They have longer half-lives and are more soluble, leading to access to cytoplasmic and nuclear substrates that are not in the proximity of cellular membranes [7]. The major form in tumorigenesis was reported to be CDK5/p25 [8], which was implicated in cell cycle regulation [9] and in the phosphorylation of the androgen receptor in prostate cancer [10]. CDK5/p25 was also involved in cancer cell migration, but one of the most interesting roles of CDK5 in oncogenesis is its potential implication in DNA repair and drug resistance [10]. It has been reported that CDK5 was activated in cancer cells or tissues exposed to conventional DNA-damaging therapies including ionizing radiation (IR) or genotoxic agents [11,12]. There, CDK5 activated cell cycle checkpoints like the checkpoint-activating kinase Ataxia-telangiectasia mutated (ATM). CDK5 has also been reported to be highly expressed in glioblastoma [13] possibly due to its location on chromosome 7, which is one of the most frequent sites of copy number increase in GBM. However, it is very rarely mutated, suggesting its stability which is crucial to be used as a therapeutic target. Taken together, CDK5 appears to be a very promising target in oncology, particularly for treating glioblastoma where irradiation and genotoxic agents are used as first line treatment. However, because of its high expression in the brain, the direct and non-specific inhibition of CDK5 could be associated with severe toxicity for patients, therefore the identification of a more specific targeted therapy was required. 

In this context, P5 is a small peptide developed at the National Institutes of Health (NIH), comprising of 24 amino acids, which specifically inhibits the tumor-related CDK5/p25 activity without affecting the normal endogenous CDK5/p35 form, and respecting physiological activity of CDK5 [14,15]. This peptide was modified with a transactivator of transcription (Tat) peptide conjugated at the C-terminus (TP5) to facilitate passage through the blood–brain barrier [14,15]. In this study, we evaluated the in vitro and in vivo applications of TP5 against glioblastoma. We showed that TP5 decreased glioblastoma cell viability and tumor growth by blocking cell cycle and increasing apoptosis through the inhibition of ATM phosphorylation. We also showed that TP5 acted synergistically with radiotherapy and temozolomide by impairing DNA damage repair.

## 2. Results

### 2.1. TP5 Decreases CDK5 Activity but Not CDK5 Expression and Can Penetrate the Cells 

TP5 was expected to specifically inhibit the activity of pathologic CDK5/P25 complex (Figure 1A,B) [16,17]. In order to determine if TP5 could reach its target into the cells, we evaluated the capacity of TP5 to decrease the activity of CDK5 using immuno-precipitation and CDK5 activity assay on two GBM cell lines (U251 and LN229). In parallel, we analyzed the impact of TP5 on CDK5 and its physiological activator p35 protein expression using Western blot. We observed that TP5 was able to decrease CDK5 activity in a dose-dependent manner with no impact on CDK5 and p35 expression (Figure 1C,D), as expected by the mechanism of action of TP5 (Figure 1A,B). We also treated these two cell lines (U251 and LN229) with TPF5, in which FITC fluorescence was conjugated to TP5 at the N-terminus and observed a homogenous signal of TPF5 in the cell cytoplasm and nucleus (Figure 1E,F). Taken together, these results confirm that TP5 is able to penetrate the cells and inhibit the CDK5 activity, without affecting the physiological expression of CDK5 and p35.

### 2.2. TP5 Alone Decreases Glioblastoma Cell Viability In Vitro by Inducing Cell Apoptosis

In order to analyze the in vitro efficacy of TP5, we used four different glioblastoma cell lines: three with *TP53* mutations (U251, LN229, T98G) and one with *TP53* wild type (U87). TP5 alone decreased the cell viability in a dose-dependent manner (Figure 2A) in all cell lines independently of *TP53* status. IC50 concentrations ranged from 17.2 µM to 52.9 µM (Figure 2A). TP5 also significantly decreased the colony number in clonogenic assay in a dose-dependent manner (Figure 2B and Appendix A). 

To understand the impact of TP5 on cell survival, we analyzed the early and late apoptosis of cells under treatment, using flow-cytometry and Western blot. We observed that TP5 increased both early and late apoptosis using PI and Annexin V staining (Figure 2C,D). Western blot also confirmed increased protein level of cleaved-PARP and cleaved-caspase 3 (Figure 2C,D). Similar effects of TP5 were also observed in T98G cells (Appendix A). Taken together, these results suggest that TP5 alone decreases cell viability by increasing the early and late apoptosis of glioblastoma cell lines.

### 2.3. TP5 Alone Affects Cell Cycle by Impairing DNA Repair

To understand the increased apoptotic rate of cells under TP5 treatment, we analyzed DNA damage and cell cycle change. Immunofluorescence staining and Western blotting showed that TP5 significantly increased the protein level of γH2A.X in a dose-dependent manner in U251 and LN229 cells (Figure 3A–C). These increased DNA damages were related to a decreased level of phosphorylated ATM (Figure 3C). Phosphorylated ATR seemed not affected by TP5 treatment as well as its main downstream effector Chk1 (Appendix A). Consequently, we observed that TP5 treatment caused G1 arrest with reduced fraction of cells in the S and G2/M phases in U251 cells and T98G cells (Figure 3D, Appendix A). Increased γH2A.X and decreased p-ATM were also confirmed in U87 cells after TP5 treatment (Appendix A). Taken together, these results suggest that TP5 treatment increases DNA damages by inhibiting the phosphorylation of ATM and impairing the DNA damage repair machinery in GBM cells.

### 2.4. TP5 Acts Synergistically with Chemotherapy and Radiotherapy by Increasing the DNA Damage Caused by Temozolomide or Irradiation through Reducing pATM

Because standard of care for glioblastoma patients was composed of two DNA-damaging therapies, chemotherapy (temozolomide [TMZ]) and radiotherapy [IR], and because the main protein implicated in DNA repair and treatment resistance is pATM [phosphorylated Ataxia Telangiectasia Mutated] which was decreased by TP5, we then analyzed the potential synergistic effect of TP5 with TMZ and IR. First, we showed that treatment of TMZ or IR significantly increased the CDK5 activity, which was impaired by the combination with TP5 (Figure 4A, Appendix A). Moreover, we demonstrated that the addition of TP5 to TMZ or IR significantly increased the protein level of γH2A.X, suggesting accumulation of DNA damage (Figure 4B,C, Appendix A). These increases in DNA-damage were due to the impaired phosphorylation of ATM by TP5. Indeed, the use of TMZ or IR increased the phosphorylation of ATM, indicating activation of DNA-damage repair pathways. However, the increased pATM was impaired by the addition of TP5 (Figure 4B, Appendix A). Then, we analyzed the synergistic effects of TP5 with serial doses of TMZ or IR using clonogenic assay. Combination of TMZ or IR with TP5 caused a dramatic reduction in colony formation (Figure 4D, Appendix A). Finally, the addition of TP5 to IR or TMZ also impaired the G2/M arrest induced by IR or TMZ (Figure 4E,F, Appendix A), leading to the accumulation of cells in the G1 phase. Taken together, our results suggest that TP5 acts synergistically with TMZ and IR by reducing the ability of tumor cells to repair DNA-damage caused by TMZ or IR, through the inhibition of ATM phosphorylation.

### 2.5. TP5 Crosses the Blood–Brain Barrier In Vivo and Suppresses GBM Tumor Growth In Vivo

To evaluate the anti-tumor efficacy of TP5, we performed a dose-escalation experiment using orthotopic U251-Luc tumor model in NSG mice. We randomized the tumor bearing mice into 4 groups: vehicle control group (H_2_O), scrambled peptide group, TP5 group 1 (100 mg/kg, every other day) and TP5 group 2 (300 mg/kg, every other day). The tumor volume evaluated by bioluminescent imaging showed slower tumor growth in both TP5 group 1 and 2, compared to the vehicle control group or scrambled group (Figure 5A–C). We then used the fluorescently labelled TPF5 to analyze the drug diffusion into the brain tumor and observed an efficient diffusion of the fluorescent peptide into the tumor (Figure 5D). We also confirmed that TPF5 (100 mg/kg) efficiently penetrated into tumor-free normal mouse brain (Appendix A). The tolerance of TP5 treatment was favorable. The blood count and biochemistry evaluation at day 18 of treatment showed no hematologic or metabolic toxicity (Figure 5E,F). No body weight loss was observed during treatment (Figure 5G). The pathological examination of mouse organs (heart, lung, liver, kidney, colon, peritoneum) showed no tissue toxicity (Figure 5H) except two localized peritoneum inflammation in the 300 mg/kg group. Based on these results, we chose 100 mg/kg as the final dose for further test. These results suggest that TP5 alone can reach the brain tumor and suppress the tumor growth with no safety concern.

### 2.6. TP5 Increases Mouse Survival Alone or in Combination with Irradiation or Temozolomide

Finally, we evaluated the impact of the combination of TP5 and TMZ or IR on orthotopic U251-Luc tumor model in nude mice. We randomized the tumor bearing mice into six groups: control (H_2_O), TP5 alone, TMZ alone, TMZ and TP5, IR alone, IR and TP5. TP5 alone significantly reduced the tumor volume compared to control group (Figure 6A,B). TP5 also increased mouse survival alone or in association with IR or TMZ (Figure 6C,D). The body weight of the treated mice was stable during the whole treatment process, suggesting that these combined treatments were well tolerated (Appendix A). These results confirmed that TP5 treatment suppressed tumor growth and combination of TP5 with standard treatments increased mouse survival.

## 3. Discussion

DNA-damage accumulation is one of the major mechanisms of action for anti-tumor treatments such as cytotoxic agents and irradiation. In the present study, we showed that the use of TP5 was able to specifically decrease the activity of CDK5 in glioblastoma cells and induce cell apoptosis. TP5 treatment decreased phosphorylation of ATM, which disrupted cell cycle and DNA damage repair. TP5 was synergistic with first-line treatment options (radiotherapy and temozolomide) by accumulating more DNA-damage. Testing in mouse xenograft models showed that TP5 alone or in combination with TMZ or IR decreased tumor volume and increased mouse survival with a favorable toxicity profile, suggesting that TP5 may be a promising therapy for glioblastoma. 

This is important as there is a critically unmet need for effective therapies for primary brain tumors. The development of anti-angiogenic failed to increase the overall survival of patients [3,4,5], and current immunotherapy approaches seem to be inactive [18]. Despite promising early results, innovative targeted therapy coupling specific monoclonal antibody-targeting EGFR with a cytotoxic agent [19] failed to increase patient survival in international prospective phase III trial. 

CDK5 appears to be an interesting and relevant target in oncology as previous preclinical studies suggested that CDK5 was integral in different tumor pathways, including tumor proliferation, migration, angiogenesis, DNA-damaging agent resistance and anti-tumor immunity [10,20]. Historically, CDK5 was not considered to be implicated in cell cycle regulation because of its predominant expression in non-dividing neurons. However, more recently, in pathological conditions such as cancer, CDK5 was reported to phosphorylate the retinoblastoma protein (Rb), promoting cancer cell proliferation [21]. Moreover, CDK5 was implicated in the phosphorylation of STAT3, leading to the deregulation of some cell cycle genes like cyclin D1 [22]. Finally, increasing evidence indicates that CDK5 contributes to the initiation of the DNA damage response and DNA repair. Upon exposure to environmental stress such as chemotherapy or irradiation, these mechanisms are activated and ensure maintenance of genome integrity during cell division and the survival of cancer cells [23,24,25,26]. CDK5 is one of the early activated proteins after exposure to conventional DNA-damaging agents, which we also observed in glioblastoma cells. CDK5 activation contributes to DNA repair by activating checkpoint proteins, which is the first step of DNA repair. Notably, CDK5 was reported to phosphorylate the checkpoint-activating kinase ATM to activate its kinase activity. ATM and ATR are the major signal transducers at very early stage after DNA damage. The ATM and ATR signaling cascades are the major responses to DNA damage and act thereby on different axes. ATR activation is driven by single strand breaks formed as a result of stalled replication forks, whereas ATM is the main initiator of response to double strand breaks resulting from ionizing radiation and other types of DNA damage [27]. Once activated, ATM and ATR phosphorylate a host of substrates, initiating a cascade that results in cell cycle arrest and DNA repair [28,29]. The results in our study and the literature review confirmed that CDK5 is a specific regulator of ATM (not ATR) [30]. Indeed, Ehrlich and colleagues reported that CDK5 activity blockade in hepatocellular carcinoma cell lines prevented ATM phosphorylation and then blocked the initiation of the DNA damage response, including the G2/M arrest [25]. In summary, these reports suggested that CDK5 may play important roles in resistance to DNA-damaging anti-cancer-therapies such as chemotherapy, IR or PARP inhibitors. The relevant combination of CDK5 inhibitors with conventional DNA-damaging agents may improve anti-cancer efficacy. In the present study, we confirmed that IR and temozolomide increased the CDK5 activity and that TP5 alone or in combination increased DNA-damage on glioblastoma cells and impaired their ability to repair DNA damage by reducing ATM phosphorylation, which is consistent with previous observations on other cancer cell lines. 

CDK5 appears to be a relevant target for brain tumors because it is highly expressed in the brain and in primary brain tumors but is very rarely mutated, conferring a stable activity and so a potential uniform sensibility to its inhibitor [13]. However, to the best of our knowledge, CDK5 was never associated with DNA-damage repair but was reported to be involved in the phosphorylation of PIKE-A [31]. Liu and colleagues showed that CDK5 directly phosphorylated PIKE-A in its GTPase domain on glioblastoma cells, leading to the activation of its downstream effector Akt that mediated migration and invasion of human glioblastoma. These results underscore another complementary implication of CDK5 in glioblastoma characterized by a diffuse invasiveness. More recently, the inhibition of CDK5 was reported to prevent glioma stem cell (GSC) self-renewal in vitro and in xenografted tumors [32]. Mukherjee and colleagues reported that CDK5 was a promoter of GSC self-renewal through its deregulation of asymmetric cell division, at least in part through its PKA/cyclic AMP-independent phosphorylation and activation of CREB1. Antagonizing CDK5 suppressed both self-renewal of GSC and glioma growth. Finally, in medulloblastoma, CDK5 was reported to help the cancer evade immune elimination by up-regulating the interferon-γ-induced PD-L1 expression [20]. Taken together, CDK5 seems to be a corner stone of primary brain tumor growth and survival, increasing interest in drug-development to target this molecule. Therefore, TP5 may have several advantages including that the selective inhibition of CDK5 activation by p25 preserves the physiological role of CDK5, reducing the side effects expected with other CDK5 inhibitors like roscovitine. Moreover, the modification of this peptide allowing its passage through the blood–brain barrier is another interesting advantage of this peptide that we were able to observe in vivo. 

## 4. Materials and Methods

### 4.1. Reagents and Compounds

Antibodies for γH2A.X (Ser139; #9718), CDK5 (#2506), β-actin (#3700), Cleaved Caspase-3 (Asp175; #9664), Cleaved PARP (Asp214; #5625), p-ATM (Ser1981; #5883), p-ATR (Ser428; #2853), Chk1 (#2360) and p-Chk1 (Ser345; #2348) were purchased from Cell Signaling Technology (Danvers, MA, USA). Antibody for phospho-H2A.X used for immunofluorescence staining was purchased from Millipore (#05-636).

TMZ was purchased from Sigma-Aldrich. TP5/TFP5 and the scrambled peptide were synthesized by Genscript (Piscataway, NJ, USA).

### 4.2. Cell Line Preparation

The U87, T98G and LN229 were purchased from the ATCC. U251 was purchased from Sigma-Aldrich. The U87, U251, LN229 and T98G were cultured in DMEM medium plus 10% FBS and 1% penicillin/streptomycin. 

### 4.3. Cell Viability Assay

Cell viability was assessed using the cell counting kit-8 (CCK-8, Dojindo, Japan) according to the manufacturer’s instruction. Briefly, 1000 cells per well were seeded in 96-well tissue culture plates. After 24 h incubation, cells were treated (control or TP5) for 72 h. Then, 10 µL of CCK-8 was added to each well and incubated for 1 h. The results were measured by the absorbance at 450 nm using Synergy H1 microplate reader (BioTek, Winooski, VT, USA).

### 4.4. Colony Formation Assay

The clonogenic assay was performed by seeding 600 cells per well in 6-well tissue culture dishes. After 24 h incubation, cells were treated (control, TP5, TMZ and irradiation) and then incubated for 10 days. The colonies were stained with 0.5% crystal violet in 20% methanol. Colonies containing greater than 20 cells were counted.

### 4.5. Immuno-Precipitation and CDK5 Activity Assay

Immuno-precipitation and CDK5 kinase activity were performed as described previously [33]. Briefly, the protein G (+) A-agarose beads were washed three times with tris-buffered saline (TBS) and incubated with Cdk5 antibody (1–2 µg/500 µg of protein lysate, Santa Cruz Biotechnology, Santa Cruz, CA, USA) for 1 h at room temperature with gentle mixing. The beads were centrifuged and washed three times with TBS and then resuspended in TBS. The protein lysates from the cells were incubated with antibody conjugated beads for overnight (O/N) at 4 °C on a rotating wheel. The beads were subsequently centrifuged and washed two times with TBS, one time with 1× kinase buffer and resuspended in kinase buffer (50 mM MOPS pH 7.2 12.5 mM B-glycerol-phosphate, 5 mM MgCl2, 5 mM EGTA, and 2 mM EDTA). The immunoprecipitated beads were used as an enzyme source for the kinase activity. For the kinase assay, a total volume of 50 μL of kinase assay mixture was used, containing 50 mM MOPS pH 7.2,12.5 mM b-glycerol-phosphate, 5 mM MgCl2, 5 mM EGTA, 2 mM EDTA, 10 μg of histone H1, and 20 μL of Cdk5 immunoprecipitates. The phosphorylation reaction was initiated by the addition of 0.1 mM [γ-32P] ATP and incubated at 30 °C for one hour. The reaction was stopped by the addition of the Laemmli sample buffer. The reaction mixture was heated for five minutes at 90 °C and electrophoresed on a 4–20% SDS-PAGE gel stained with Coomassie blue, and then dried and exposed overnight for the detection of 32P-labeled Histone H1 by autoradiography. The films were scanned, and the bands were quantified using Image J software.

### 4.6. Immunoblot Analysis

The cell lysates were processed with RIPA buffer containing protease inhibitor cocktail (4693132001; Roche, Bâle, Suisse) and phosphatase inhibitor cocktail (4906845001; Roche) and were collected for Western blotting as describe previously [34].

### 4.7. Immunofluorescence Staining

The cells were analyzed 24 h after treatment. They were washed with PBS, fixed in 2% paraformaldehyde (PFA) and then 70% ethanol. Cells were blocked by BSA (5%) and then stained with the primary and the secondary antibodies. Xenograft tumor tissue was harvest at the time of mouse euthanasia for TPF5 analysis. Images were taken under Zeiss LSM710 confocal microscope. 

### 4.8. Apoptosis Assay

Cell apoptosis was analyzed by APC Annexin V staining according to the manufacturer’s instruction (BD Biosciences, San Jose, CA, USA, 550474). Briefly, cells were harvested after indicated treatment and washed with PBS and Annexin V binding buffer (BD Biosciences, 556454). The cells were incubated with Annexin V-APC in 100 µL Annexin V binding buffer at room temperature for 10 min. Next, 400 µL Annexin V binding buffer containing 1 µL PI was added to the cells. After 15 min incubation on ice, the stained cells were analyzed within one hour by flow cytometry (BD LSRFortessa SORP, San Jose, CA, USA).

### 4.9. Cell Cycle Analysis

For cell cycle analysis, one million cells were washed twice with PBS and fixed in 70% ethanol. Cells were then treated with 500 µL of FxCycle PI/RNase Staining Solution, incubated for 30 min and analyzed by flow cytometry (Life Technologies, Carlsbad, CA, USA).

### 4.10. Intracranial GBM Model, Bioluminescence Image, and Treatment

An orthotopic GBM mouse model was used for the evaluation of treatment efficacies of TP5, TMZ, irradiation and combined treatment in vivo according to an approved animal study protocol by the NCI-Animal Use and Care Committee (animal protocol: NOB-019). U251-Luc cells were injected stereotactically into the striatum of nude mice (6–8 weeks old; Charles River Frederick Research Model Facility) using a stereotactic device (coordinates, 2 mm anterior and 2 mm lateral from bregma, and 2.5 mm depth from the dura). Tumor growth was monitored by luciferase expression using the PerkinElmer IVIS Spectrum until the first mouse reached endpoints. Mice were imaged 15 min after intraperitoneal injection of luciferin at 150 mg/kg. 

For the first dose-escalation experiment, mice were treated by TP5 (IP. 100 mg/kg or 300 mg/kg every 2 days), scrambled peptide (IP. 300 mg/kg every 2 days), or vehicle control (H_2_O: 200 µL every 2 days), respectively.

For the combination therapy experiment, mice were divided into 6 groups: control, TP5 alone, TMZ alone, irradiation alone, TMZ and TP5, irradiation and TP5. The mice were treated with TP5 (IP. 100 mg/kg day 1 and then 50mg/kg every 2 days), and/or TMZ (p.o. gavage daily for 5 days, 5 mg/kg), and/or vehicle control (H_2_O: 200 µL), and/or whole brain irradiation (day1: 6Gy).

### 4.11. Blood Test and Organ Examination

Mouse facial vein blood was collected in K2EDTA tubes (BD Microtainer, Franklin Lakes, NJ, USA) and sent to the Department of Laboratory Medicine at Clinical Center of NIH for complete blood count (CBC) and animal screen chemistry test. When the animals reached endpoints, they were euthanized by carbon dioxide narcosis and cervical dislocation. The brains and other major organs were dissected for histopathologic examination.

### 4.12. Statistical Analysis

Statistical analyses were performed using a GraphPad Prism software (Version 6.05, GraphPad Software, Inc., San Diego, CA, USA) and SPSS v22 software. They were considered significant at * *p* < 0.05; ** *p* < 0.01; *** *p* < 0.001. Data are shown as mean ± SEM. One-way ANOVA or independent and dependent Student’s t-test were used for statistical analysis. Mouse survival was analyzed using Kaplan–Meier and Log-rank tests.

## 5. Conclusions

In conclusion, TP5 appears to be a promising therapy to use alone or in combination with radio-chemotherapy for glioblastoma patients. Further investigations will evaluate its potential applications in other cancer types before translation to patients in early clinical trials.

## Figures and Tables

**Figure 1 cancers-12-01935-f001:**
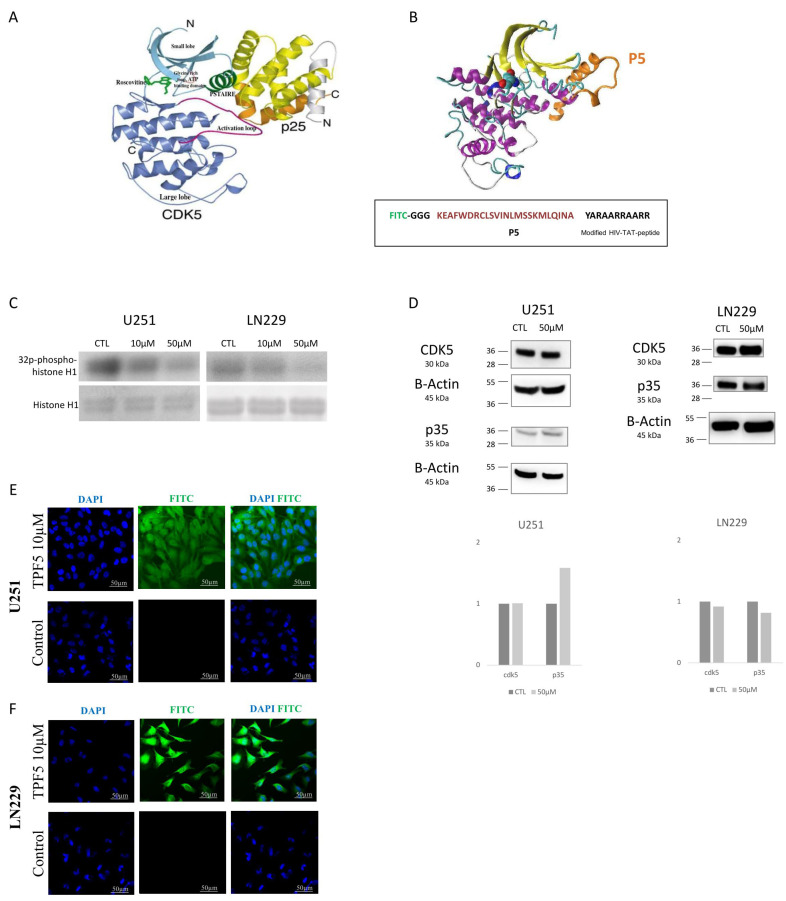
TPF5 structure and activity. (**A**) Crystal structure of the pathologic CDK5/P25 complex. Important structural elements for CDK5 activity are indicated in the figure [16]. (**B**) Possible inhibitory CDK5-p5 binding mode. More refined binding modes were obtained in [17]. (**C**) CDK5 activity is indicated by phosphorylated histone 1 in U251 and LN229 cells treated by TP5 at indicated concentrations (CTL: control). (**D**) Western blot of CDK5 and p35 in U251 and LN229 cells treated by TP5 at indicated concentrations. (**E**) TPF5 (10 µM) efficiently penetrated into U251 cells. (**F**) TPF5 (10 µM) efficiently penetrated into LN229 cells.

**Figure 2 cancers-12-01935-f002:**
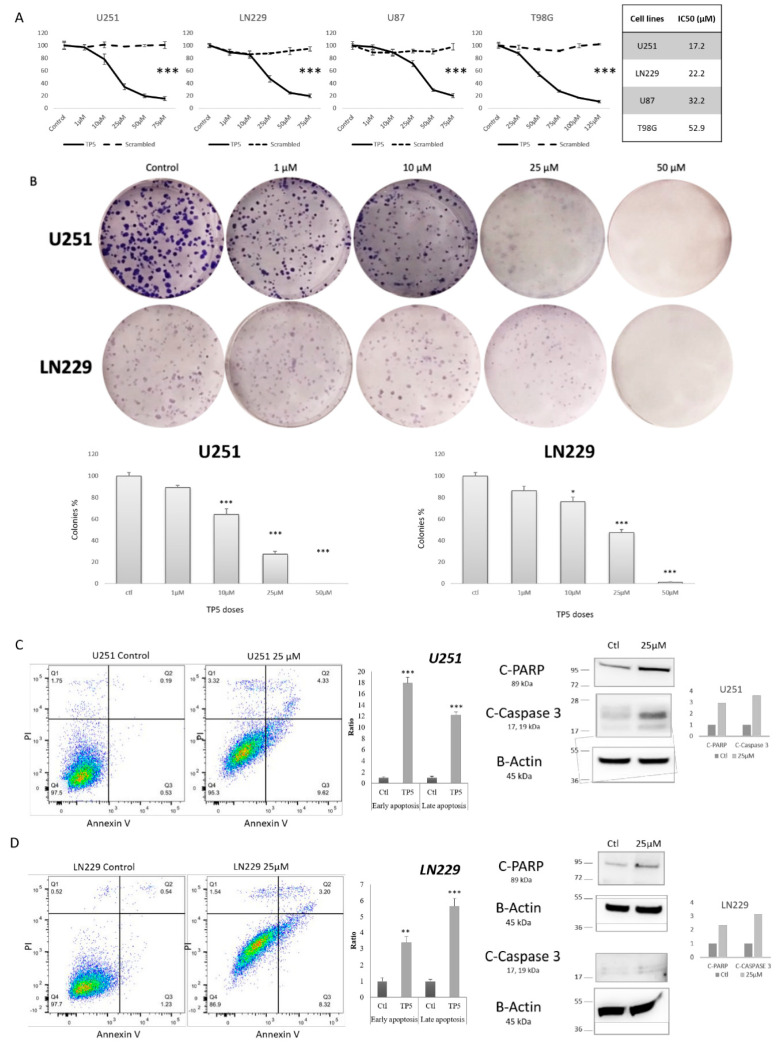
TP5 decreases cell viability by inducing cell apoptosis. (**A**) Viability of U251, LN229, U87 and T98G cells treated by indicated concentrations of TP5 or scrambled peptide for 72 h is shown (*** *p* > 0.001). Right table: IC50 concentrations. (**B**) Clonogenic growth of U251 (top panels) and LN229 (bottom panels) cells is shown. The bar graphs display quantification of colonies under treatment at indicated concentrations (*N* = 3) (* *p* < 0.05; *** *p* < 0.001). (**C**) Apoptosis analysis by Annexin V/Propidium Iodide (PI) staining. Left panel: Representative flow cytometry dot plot graphs of annexin V and PI in U251 cells are shown. The bar graph displays the quantification of early (Q3) and late (Q2) apoptotic cells after treatment by TP5 (25 µM) in U251 cells (*N* = 4) (*** *p* < 0.001). Right panel shows protein level of cleaved PARP and cleaved Caspase 3 in U251 cells after 24 h of treatment by TP5 (25 µM). (**D**) Apoptosis analysis in LN229 cells. Left panel: Representative flow cytometry dot plot graphs of annexin V and PI in LN229 cells are shown. The bar graph displays the quantification of early and late apoptotic cells after treatment by TP5 (25 µM) in LN229 cells (*N* = 4; ** *p* < 0.01; *** *p* < 0.001). Right panel shows protein level of cleaved PARP and cleaved Caspase 3 in LN229 cells after 24 h of treatment by TP5 (25 µM).

**Figure 3 cancers-12-01935-f003:**
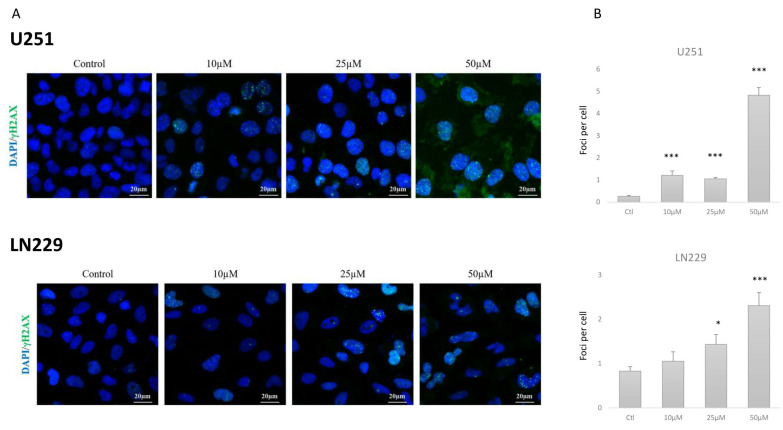
TP5 affects cell cycle by impairing DNA damage repair. (**A**) Immunofluorescent staining of γH2A.X (green) in U251 and LN229 cells after treatment of TP5 at indicated concentrations. (**B**) The bar graph displays the quantification of γH2A.X foci per U251 and LN229 cell treated at indicated concentrations (ten random fields of view for each group, on 3 distinct experiments; * *p* < 0.05, *** *p* < 0.001). (**C**) Western blot of phosphorylated ATM and γH2A.X in U251 cells (left panels) and LN229 cells (right panels) treated by TP5 at indicated concentrations. (**D**) Cell cycle analysis in U251 cells treated by TP5 at 25 µM for 48 h. The graph bar displays the cell cycle phase quantification (*N* = 3; * *p* < 0.05; ** *p* < 0.01).

**Figure 4 cancers-12-01935-f004:**
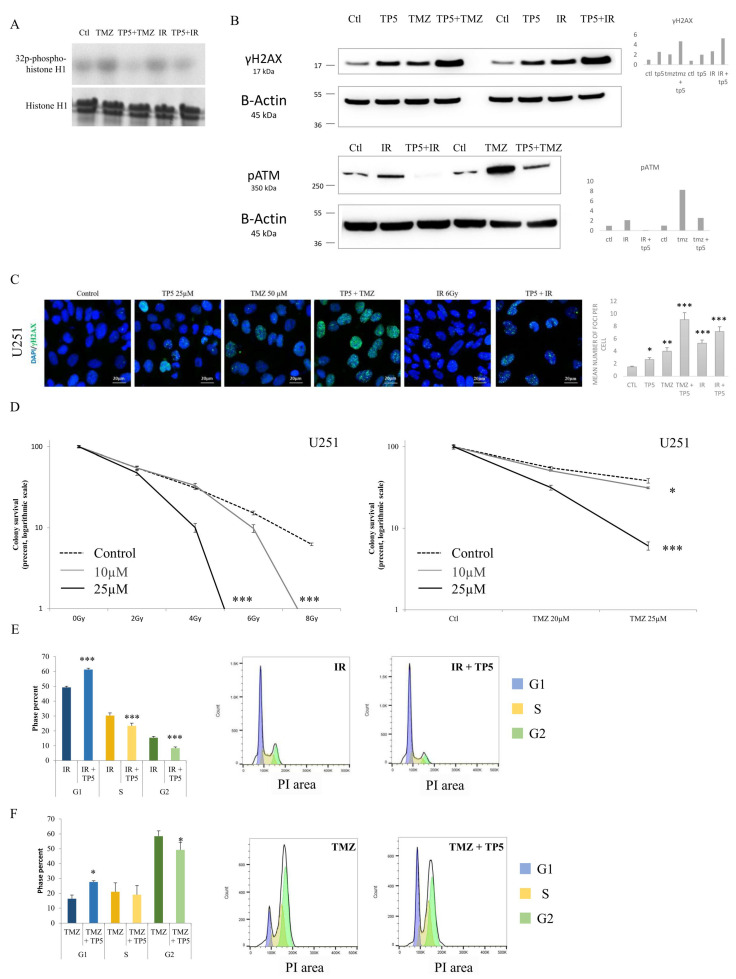
TP5 acts synergistically with chemotherapy and radiotherapy. (**A**) CDK5 activity is shown by phosphorylated histone 1 in U251 cells treated as indicated (Ctl: control; TMZ: temozolomide 50 µM; TP5: 25 µM; IR: irradiation 6 Gy). (**B**) The protein levels of γH2A.X and phosphorylated ATM are shown by Western blot in U251 cells treated as indicated (Ctl: control; TMZ: temozolomide 50 µM; TP5: 25 µM; IR: irradiation 6 Gy). (**C**) Left panel: Immunofluorescent staining of γH2A.X (green) in U251 cells after treatment as indicated. Right panel: the bar graph displays the quantification of γH2A.X foci per cell treated as indicated (ten random fields of view for each group, on 3 distinct experiments; * *p* < 0.05; ** *p* < 0.01; *** *p* < 0.001). (**D**) Synergistic interactions between TP5 and irradiation (left graph) or chemotherapy (TMZ: temozolomide, right panel) are shown by clonogenic growth of U251 cells treated as indicated (TP5 at 10 and 25µM; irradiation from 0 to 8 Gy; TMZ at 20 and 25 µM) (*N* = 3; * *p* < 0.05; *** *p* < 0.001). (**E**) Cell cycle analysis is shown for U251 cells treated by TP5 (25 µM) and irradiation (6 Gy) for 48 h. The graph bar displays the cell cycle phase quantification (*N* = 4; *** *p* < 0.001). (**F**) Cell cycle analysis is shown for U251 cells treated by TP5 (25 µM) and TMZ (50 µM) for 48 h. The graph bar displays the cell cycle phase quantification (*N* = 3; * *p* < 0.05).

**Figure 5 cancers-12-01935-f005:**
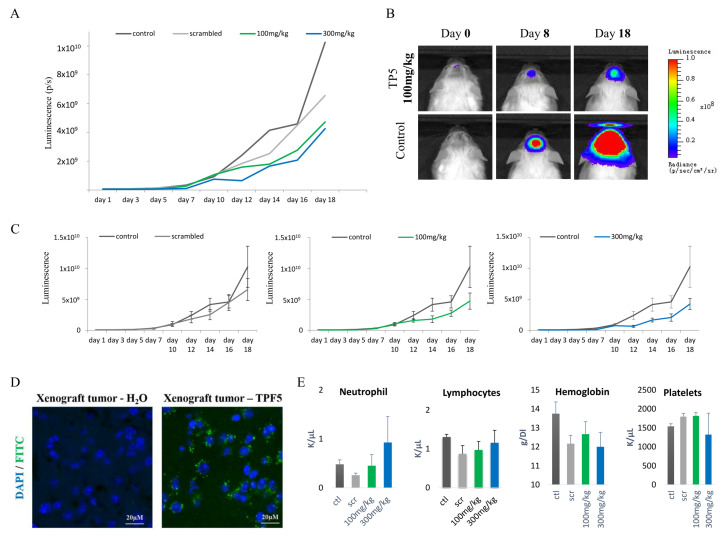
TP5 crosses the blood–brain barrier in vivo, suppresses GBM tumor growth in vivo and shows no systemic toxicity. (**A**) Tumor volume over time is shown for the 4 groups (Control *N* = 4; Scrambled *N* = 3; TP5 100 mg/kg *N* = 4; TP5 300 mg/kg *N* = 4). (**B**) Representative bioluminescent images of control and TP5 (100 mg/kg) groups on day 0 (first day of treatment), day 8 and day 18 (last day of experiment). (**C**) Tumor volume (luminescence signal) over time is shown for each group: control and scrambled peptide (left panel); control and TP5 at 100 mg/kg (middle panel); control and TP5 at 300 mg/kg (right panel). (**D**) TPF5 signal (green) in the xenograft intra-cranial tumor area is shown. (**E**) Blood hematologic count of neutrophils, lymphocytes, hemoglobin and platelets at day 18 of treatment is shown in the four groups (ctl: control *N* = 4; scr: scrambled *N* = 3; TP5 at 100 mg/kg *N* = 4 and 300 mg/kg *N* = 4). No significant difference was observed. (**F**) Blood concentration of AST, triglycerides, cholesterol, urea, amylase and glucose at day 18 of treatment is shown in the four groups (ctl: control *N* = 4; scr: scrambled *N* = 3; TP5 at 100 mg/kg *N* = 4 and 300 mg/kg *N* = 4). No significant difference was observed. (**G**) Mouse weight over time is shown from the first day of treatment until the end of experiment (day 18). (**H**) Pathological examination of mouse organs is shown: 1- lung; 2- heart; 3-kidney; 4- liver; 5-colon; 6-peritoneum.

**Figure 6 cancers-12-01935-f006:**
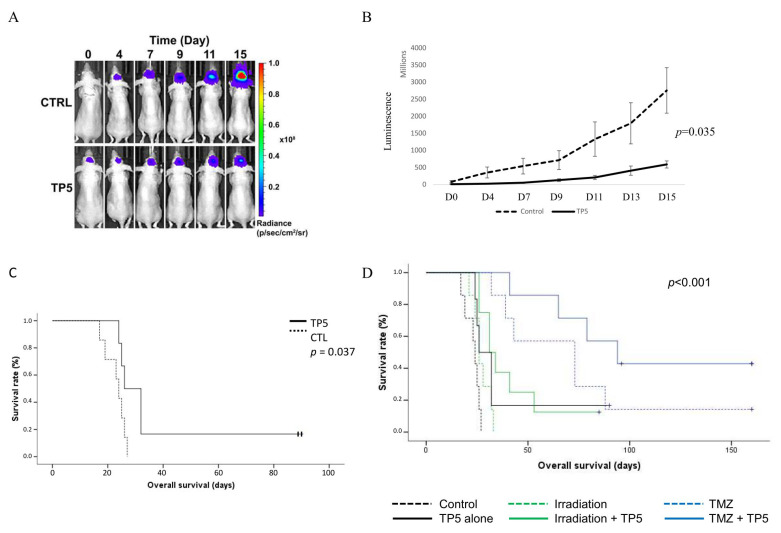
TP5 increases mouse survival alone or in combination with irradiation or temozolomide. (**A**) Representative bioluminescent imaging of control and TP5 only groups on day 0, 4, 7, 9, 11, 15. (**B**) Tumor volume of control and TP5 only groups over time (luminescent signal) is shown from the first day of treatment to the day 15 (first death in the control group). (**C**) Survival curve of control and TP5 only group. (**D**) Mouse overall survival according to the treatment groups is shown: control (*N* = 7), TP5 alone (100 mg/kg day 1 and then 50 mg/kg every 2 days, *N* = 6), Irradiation 6 Gy (*N* = 7), TMZ (temozolomide 5 mg/kg for 5 days, *N* = 7), Irradiation and TP5 (*N* = 8), TMZ and TP5 (*N* = 7).

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
