# Peer review of "TP5, a Peptide Inhibitor of Aberrant and Hyperactive CDK5/p25: A Novel Therapeutic Approach against Glioblastoma"

_cancers, 2020, doi:10.3390/cancers12071935_

Round 1
Reviewer 1 Report
The authors covered most of the concerns how ever I feel that fig 3 c , will be incomplete without panel of pchk1 and total chk1.Th
Thank you
Reviewer 2 Report
The authors addressed all the comments. The manuscript can be accepted. However, the authors should use primary GBM cells for future studies.
Reviewer 3 Report
Ms# cancers-826148
TP5, a peptide inhibitor of aberrant and hyperactive CDK5/p25: a novel therapeutic approach against glioblastoma.
Emeline Tabouret, Herui Wang, Niranjana Amin, Jinkyu Jung, Romain Appay, Jing Cui, Qi Song, Antonio Cardone, Deric M. Park, Mark R. Gilbert, Harish Pant and Zhengping Zhuang
Comments:
The authors answered the comments pointed out in the review and modified the paper accordingly the suggestions.
This manuscript is a resubmission of an earlier submission. The following is a list of the peer review reports and author responses from that submission.
Round 1
Reviewer 1 Report
The article “TP5, a peptide inhibitor of aberrant and hyperactive 2 CDK5/p25: a novel therapeutic approach against glioblastoma” by Emeline et. al; is a good attempt to try the efficacy of a novel peptide against glioblastoma. Inhibiting CDK5 activity is very a novel approach in glioblastoma and most interesting aspect is that TP5 will cross the blood brain barrier. The authors argue that TP5, the peptide by inhibiting CDK5, impairs ATM phosphorylation, thereby weakening DNA damage repair and cause apoptosis.
- Even though ATM phosphorylation is inhibited by TP5, still other DNA damage repair pathway may be still active. The authors may address this also.
- In Fig 3, both the cells show high endogenous level of ATM phosphorylation, which is concerning indicated the cells are either highly stressed or have high base level repair. To make it more logic, the western blot panel should contain, total ATM, total Chk2, and phospho-chk2 blots too.
- The authors must show the time dependent assay of TP5 induced γ- H2AX foci and WB to show the degree of DNA damage repair inhibition by TP5.
- In Fig 4C, as the authors mentioned Temozolomide is given for 24 hours, which persistently induce DNA damage and accumulate the γ-H2AX foci, as it won’t get repaired, the combination won’t show significant increase in γ- H2AX foci. It probably an experimental error. To confirm this, a 48-hour experiment is warranted.
- Also, in Fig 4F, cell cycle analysis is recommended for 48hours
- Fig 5A and 5B doesn’t match, as 5A show a radiance of 10 9, on 18th day, 5B show 10 8 only. Also, from the tumor luminescence trend, one would expect, there is no significant difference between vehicle and TP5 treated group.
- In Fig 6, the authors may include one more panel with the individual luminescence of the animals to have a stronger picture about the tumor size. From the survival graph, it is not clear that the TP5 alone mice which survived longer may be an outliner.
- In addition, may be also noted that the cell lines used in this study, U251 and LN 229 are p53 mutant, authors may use another cell line with a p53 WT status to demonstrate that the effect is p53 independent, as there are reports that ATM inhibitor have differential effect on p53 status.
Reviewer 2 Report
Emeline Tabouret et al. examined the efficacy of selective inhibition of CDK5 in glioblastoma by TP5. They found that TP5 decreased the activity but not the expression of CDK5 and p35. TP5 alone impaired cell viability and colony formation of glioblastoma cell lines and induced apoptosis. TP5 increased DNA damage by inhibiting the phosphorylation of ATM, leading to G1 arrest. Whereas CDK5 activity is increased by DNA-damaging agents such as temozolomide and irradiation, TP5 was synergistic with either temozolomide or irradiation due to an accumulation of DNA damage. Concomitant use of TP5 and either temozolomide or irradiation reduced the phosphorylation of ATM, increased DNA damage and inhibited the G2/M arrest induced by temozolomide or irradiation. TP5 alone suppressed the tumor growth of orthotopic glioblastoma mouse-model. The development of new therapeutic strategies is a critical need in glioblastoma and this study provides evidence that TP5 is a novel therapeutic approach against glioblastoma. However, there are some specific comments:
- Figure 1D, the authors may also show dose 10uM in Figure 1D and 50uM in Figure 1E and F.
- Page 3, line 104, should be Figure 2B, not 2C.
- The authors tested TPF5 using dose 10uM and 50uM in Figure 1, why use 25uM in Figure 2 for apoptosis assay?
- In Figure 2, cell viability assay using 72 hours, however apoptosis assay using 24 hours, why the authors choose different time points? Figure 2C, how to explain c-parp is positive in Ctrl group using U251 cells? The authors should also show total PARP and total Caspase 3 expression instead of Actin. Figure 2D, the authors should also show WB result using LN229 cells.
- Figure 4, what is TMA IC50 of U251 cells? The authors should use another GBM cell model to comfirm the finding in Figure 4.
- The authors need to confirm the in vivo finding using another GBM cell model.
Reviewer 3 Report
Ms# cancers-739637
TP5, a peptide inhibitor of aberrant and hyperactive CDK5/p25: a novel therapeutic approach against glioblastoma.
Emeline Tabouret, Herui Wang, Niranjana Amin, Jinkyu Jung, Romain Appay, Jing Cui, Qi Song, Antonio Cardone, Deric M. Park, Mark R. Gilbert, Harish Pant and Zhengping Zhuang
Glioblastoma (GBM) is the most common and aggressive primary brain tumor. Although during the last years many advances have been made, its prognosis is still poor. Lately, CDK5 has appeared as a promising target for glioma treatment, in fact this kinase is aberrantly activated in GBM and its inhibition prevents glioma-initiating cell self-renewal in vitro and in xenograft tumors. Here the authors describe how the small peptide TP5 decreased GBM cell viability and tumor growth by blocking cell cycle and increasing apoptosis through the inhibition of DNA damage repair.
Comments:
The authors have used as cellular models for assessing cell viability after TP5 treatment TP53-mutated GBM cell lines U251, LN229 and T98G and the TP53 wild type GBM cell line U87. However, clonogenic assays and apoptosis analysis by Annexin V/Propidium Iodide (PI) staining were performed only on U251 and LN229. Curiously, protein level of cleaved PARP and cleaved Caspase 3 were analyzed only by western blot in U251 cells. It will be interesting to check the apoptosis induction in the different cell lines included in the study. Additionally, and although the effect of TP5 on cell viability seems to be independent of TP53 status, the authors did not check if TP5 alters cell cycle through its role on DNA damage repair in TP53-wild type GBM cell lines. Finally, the authors mention that TP5 crosses the blood-brain-barrier (BBB), nevertheless it has been described that the presence of abnormal vessels in the tumor is associated with increased permeability. Did the authors check if TP5 crosses the BBB in normal brain?
Minor comments:
Figure 2 (panel B) showing clonogenic assays for U251 and LN229 cells should be improved.
